# An Association Rule Mining Analysis of Lifestyle Behavioral Risk Factors in Cancer Survivors with High Cardiovascular Disease Risk

**DOI:** 10.3390/jpm11050366

**Published:** 2021-05-02

**Authors:** Su Jung Lee, Kathleen B. Cartmell

**Affiliations:** 1Research Institute on Nursing Science, School of Nursing, Hallym University, 1 Hallymdaehak-gil, Chuncheon-si 24252, Korea; sujunglee@hallym.ac.kr; 2Department of Public Health Sciences, Clemson University, 519 Edwards Hall, Alpha Epsilon Drive, Clemson, SC 29634, USA

**Keywords:** cancer survivor, lifestyle risk behavior, cardiovascular disease, health risk assessment, association rule mining

## Abstract

We aimed to assess which lifestyle risk behaviors have the greatest influence on the risk of cardiovascular disease in cancer survivors and which of these behaviors are most prominently clustered in cancer survivors, using logistic regression and association rule mining (ARM). We analyzed a consecutive series of 897 cancer survivors from the Korean National Health and Nutritional Exam Survey (2012–2016). Cardiovascular disease risks were assessed using the atherosclerotic cardiovascular disease score (ASCVDs). We classified participants as being in a low-risk group if their calculated ASCVDs was less than 10% and as being in a high-risk group if their score was 10% or higher. We used association rule mining to analyze patterns of lifestyle risk behaviors by ASCVDs risk group, based upon public health recommendations described in the Alameda 7 health behaviors (current smoking, heavy drinking, physical inactivity, obesity, breakfast skipping, frequent snacking, and suboptimal sleep duration). Forty-two percent of cancer survivors had a high ASCVD. Current smoking (common odds ratio, 11.19; 95% confidence interval, 3.66–34.20, *p* < 0.001) and obesity (common odds ratio, 2.67; 95% confidence interval, 1.40–5.08, *p* < 0.001) were significant predictors of high ASCVD in cancer survivors within a multivariate model. In ARM analysis, current smoking and obesity were identified as important lifestyle risk behaviors in cancer survivors. In addition, various lifestyle risk behaviors co-occurred with smoking in male cancer survivors.

## 1. Introduction

The number of cancer survivors is rapidly increasing worldwide due to improvements in early diagnosis and treatment of cancer [1]. Due to these advances in cancer diagnosis and treatment, deaths due to treatment complications and other chronic diseases are gradually increasing in cancer patients [2]. Several recent studies have reported that cancer survivors have a high risk of cardiovascular disease [3], and cancer survivors who are at risk of cardiovascular disease have a significantly higher mortality rate than those who do not have cardiovascular risk [4,5]. In particular, this disease trend is prominent among cancer survivors who have survived for over five years [6,7], and the risk of cardiovascular disease in long-term cancer survivors is actually higher than the recurrence rate of cancer [8].

There are several causes of increased cardiovascular risk in cancer survivors. First, cardiotoxic drug treatments such as anthracycline [9], paclitaxel, and trastuzumab [10], and mediastinal radiation therapy can directly damage the cardiovascular system [11]. Second, cancer is usually associated with thrombosis due to pro-inflammatory cytokines and endothelial damage, which in turn increases the risk of cardiovascular disease in cancer survivors [12]. Third, risk factors related to lifestyle, such as obesity and low physical activity, which are common among cancer survivors can increase the risk of cardiovascular disease in cancer survivors [13]. According to a Canadian Lerond report, factors that contribute to health and the development of chronic diseases are 20% attributable to genetic factors, 20% to environmental factors, 8% to medical services, and 52% to personal lifestyle factors [14]. In other words, lifestyle risks such as physical inactivity, smoking, and alcohol consumption are the main factors that contribute to chronic disease risk [15]. In particular, ischemic heart disease is mainly attributable to lifestyle risk behaviors such as lack of exercise, excessive drinking, smoking, and obesity [16]. However, most previous studies have focused on individual lifestyle risk behaviors in cancer survivors. In other words, most of the studies analyzed only selective lifestyle factors such as smoking and obesity, or only analyzed data using the simple sum of the number of lifestyle risk factors [17,18]. It is difficult to identify specific lifestyle factors that are related to each other, or that cluster together in individuals. Thus, we aimed to assess which lifestyle risk factors influence the risk of cardiovascular disease in cancer survivors and which of these factors tend to cluster together in cancer survivors using logistic regression and association rule mining (ARM).

## 2. Materials and Methods

### 2.1. Study Design and Participants

This study was conducted as a cross-sectional analysis of the fifth period (2012), the sixth period (2013–2015) and the seventh period (2016) of the Korea National Health and Nutrition Examination Survey. We conducted this research after approval of exemption from review by the Institutional Board and Ethics Committee (IRB number: HIRB-2020-022). KNHANES participants are a representative population whose records are extracted randomly by resident registration number and housing census numbers in South Korea. The components of this study consist of a health survey, a medical examination, and a nutritional survey, which are conducted by trained research staff. The health survey is conducted through personal interviews, and the examination is conducted by direct measurement. Among a total of 39,156 participants in the 2012–2016 KNHANES Survey, 1017 respondents answered yes to the question, “Have you ever been diagnosed with cancer by a doctor?”. Of these, 897 were the final subjects for this analysis, with exclusion of 120 patients who had missing clinical data.

### 2.2. Measures

#### 2.2.1. Demographic and Disease Characteristics

We included age, gender, household income level, degree of education, and marital status of the subjects. Household income was defined as total income in the family divided by the total number within a family and divided into four quartiles (lower, lower middle, upper middle, upper). Level of education was classified as less than 6 years, 7–9 years, 10–12 years, and 13 years or more. Marital status was classified as ‘living with a spouse or domestic partners’ or as ‘single, divorced or widowed’.

#### 2.2.2. Anthropometric Characteristics

Among the examination items, this study used height, weight and blood pressure, which were measured by a trained examiner. Blood pressure was measured three times in a sitting position by a trained examiner, with five-minute breaks between measurements, and the average of the three measurements was used for the blood pressure parameter. For blood tests, subjects were asked to fast for a minimum of 8 h before taking the test. Blood tests were collected, following overnight fasting of at least eight hours. Blood cholesterol and glucose status were measured by Hitachi automatic analyzer 7600 (Tokyo, Japan).

#### 2.2.3. Lifestyle Risk Behaviors

Lifestyle risk behaviors were identified based upon the Alameda County studies, which reported that individual health behaviors are highly linked to disease occurrence and death [19]. Therefore, we included current smoking, heavy drinking, physical inactivity, obesity, breakfast skipping, frequent snacking, and sub-optimal sleep duration as lifestyle risk behaviors. Smoking was defined as having smoked more than 100 cigarettes in one’s lifetime or being a current smoker [20]. Heavy drinking was defined for men as drinking 7 or more cups and for women as drinking 5 or more cups at a time at least twice a week [21]. Physical inactivity was defined as the lack of participation in either high intensity exercise (for more than 25 min once a week for more than three days) or moderate intensity exercise (for more than 30 min five times a week) [22]. Breakfast skipping was defined as not eating breakfast either that day or the day before. Frequent snacking was defined as eating between meals 3 or more times per day, whereas infrequent snacking was defined as snacking fewer than 3 times per day, regardless of type and quantity [23]. Sub-optimal sleep was defined as sleeping either less than 7 or more than 8 h per night [24].

#### 2.2.4. Cardiovascular Risk

To assess cardiovascular disease risk, we used the atherosclerotic cardiovascular disease (ASCVD) risk score, as revised in 2013 [25]. The ASCVD is a simple method for calculating the risk of being diagnosed with cardiovascular disease within ten years, calculated based upon age, total cholesterol, high-density lipoprotein cholesterol, hypertension, diabetes, systolic blood pressure, and smoking status. We classified participants as being in the low-risk group if their calculated ASCVDs was less than 10% and as being in the high-risk group if their ASCVDs was 10% or higher.

#### 2.2.5. Statistical Analysis

We compared demographic and clinical characteristics between the low and high ASCVD groups using the χ2 or t-test (or Mann–Whitney U test). We performed binary logistic regression analysis to assess which lifestyle risk behaviors were related to high ASCVD score. The association between lifestyle risk behaviors and high ASCVD were analyzed after adjustment for statistically significant demographic variables, and their association was reported as an odds ratio (OR) with 95% confidence interval (CI). Variables with *p* value less than 0.05 in the univariate model were included in the multivariate model as an ‘input’ method.

#### 2.2.6. Association Rule Mining

The ARM was first introduced by Agrawal et al. as an analysis method that quantifies the simultaneous correlations between various features in a cluster [26]. In other words, dichotomous variables are often found together within a group of interest, and among them, highly correlated variables with respect to a particular target are found with statistical parameters [27]. The a priori algorithm is the most well-known ARM algorithm and quantifies the correlation of the most frequent occurrences of items with high support, confidence, and lift, showing that each occur frequently within a given group [28]. The support for the rule (A → B) is the probability that the two behaviors occur together.
$$Support\;\left(A\to B\right)=P\left(A\cap B\right)=\frac{Number\;of\;persons\;having\;A\;and\;B}{Total\;number\;of\;persons}$$

The confidence for the rule (A → B) is the conditional probability of having B given that a patient has an A item.

The lift of the rule (A → B) is the confidence of the rule divided by the unconditional probability of the consequent (B).
$$\mathrm{Lift}\;\left(A\to B\right)=\frac{Confidence\;\left(A\to B\right)}{Support\;\left(B\right)}$$

The higher the lift value between two variables, the higher the correlation of the two variables. The left-hand side of the rule (e.g., A) is the antecedent and the right-hand side the consequent (e.g., B)

We present the statistical significance of the association rules using the chi-square test. All statistical methods were performed with R version 3.6.3 (the R foundation)

## 3. Results

A total of 897 cancer survivors were included in the final analysis. Of the 897 subjects, 358 (39.9%) were males and 539 (60.1%) were females. The mean age of subjects was 62.2 (±8.1) years, respectively. A large proportion of cancer survivors reported physical inactivity, breakfast skipping, obesity, heavy drinking, and current smoking. Breakfast skipping and frequent snacking were more frequent in low- risk groups than in high-risk groups. Current smoking was found to be twice as prevalent in the high-risk group, as compared to the low-risk group.

### 3.1. Predictors of Lifestyle Risk Behaviors That Are Associated with High ASCVD Risk

Table 1 displays the comparison of baseline characteristics and health risk behaviors between low and high ASCVD score groups. In crude comparison of demographic and laboratory parameters, age, male gender, low educational year, marital status, diabetes and cardiometabolic laboratory parameters including, blood glucose, blood pressure, and cholesterol levels were associated with high ASCVD risk.

Table 2 summarizes the logistic regression analysis of lifestyle risk behaviors in predicting high ASCVD risks in cancer survivors. In univariate models, current smoking and physical inactivity were predictors and frequent snacking was a negative predictor of high ASCVD risk. However, current smoking and obesity were significant predictors for high ASCVD in cancer survivors in the multivariate model. Physical inactivity was a positive predictor in the crude model. However, its association with the ASCVD risk was reversed in the fully adjusted model.

### 3.2. Result of Association Rule Mining

Table 3 displays the result of significant association rules for high ASCVD risk in male cancer survivors. Men with physical inactivity, optimal sleep duration, and frequent snacking was the most frequent rule (rule No.2; count, 16), and it was associated with low ASCVD risk. Among important association rules, high ASCVD risk was associated with 4 important rules in male survivors, in which all the rules were associated with current smoking. Figure 1 depicts these associations with size and color quantitatively. However, in female survivors, there were no important rules associated with high ASCVD (Table 4, Figure 2) in ARM analysis. In particular, rules of frequent snacking in female survivors (rule No 1–4) had the highest lift value in ARM analysis.

## 4. Discussion

In this study, we assessed the lifestyle risk behaviors that predict high ASCVD risk with logistic regression and identified some patterns of clustering of lifestyle risk behaviors in cancer survivors with high ASCVD risk. Current smoking and obesity were significant predictors for high ASCVD risk. Moreover, current smoking was the important behavior that influenced high ASCVD risk in male survivors within the ARM analysis, and it co-occurred with low physical activity, obesity and heavy drinking, simultaneously. Identifying important high risk clusters using ARM analysis is a useful method that can show the inter-correlation of several risk factors with small but significant associations in the same hierarchy.

According to the American Cancer Society and American Heart Association, the major modifiable factors affecting both cardiovascular disease and cancer are weight management, regular physical activity, healthy eating, drinking restrictions, and smoking, which are recommended to maintain a healthy lifestyle [29]. However, several reports on lifestyle behaviors in cancer survivors have reported that a large proportion of cancer survivors continued to smoke after the cancer diagnosis [30,31], were overweight or obese, and were significantly less likely to be physically active compared to the participants without a history of cancer diagnosis [30,31,32]. Our study also showed that a larger proportion of cancer survivors did not stop smoking, were physically inactive and obese, and had improper eating habits including skipping breakfast. In addition, 42% of cancer survivors had a 10% or higher risk of developing cardiovascular disease within 10-years. These findings reconfirm that lifestyle modification interventions such as smoking cessation are highly warranted in order to achieve optimal health outcomes among cancer patients.

In our binary logistic regression analysis results, smoking and obesity were significant predictors for high ASCVD risk in our cancer survivors. In our gender-specific ARM analyses, there was no significant association rules that related to the high ASCVD risk in female survivors. However, some patterns of co-existing multiple health risks associated with smoking have been observed in men with high risk of ASCVD. Smoking is one of the main causes of cancer, is associated with about 16% of all cancer development and 30% of all cancer deaths [33,34]. Furthermore, if smoking continues after cancer diagnosis, both all-cause mobility and cancer-specific mobility increase and are associated with increased risk of secondary cancer and complications associated with surgery, radiation therapy and chemotherapy [35,36]. From the point of view of atherosclerosis development, smoking is one of the most important preventable risks ever known. However, previous survey results showed that only 62% of patients who were smokers at the time of cancer diagnosis were instructed by doctors to quit smoking, and only 44% reported that health care providers had been trained on the dangers of smoking [37]. Therefore, our findings suggest that interventions are needed to systematically provide cancer patients with evidence-based interventions to help them to quit smoking and to address other high-risk health behaviors that contribute to cardiovascular disease risk.

In our study, physical inactivity was shown to be protective in our logistic regression analysis. However, aerobic exercise is associated with reducing chemotherapy-induced cardiotoxicity [38] and reducing the incidence of heart failure and coronary artery disease [39]. Therefore, the results of the preventive effect of physical inactivity on the ASCVD risk in our data should be evaluated cautiously, within the context the results of the logistic regression and the ARM analysis. In our subjects, the prevalence of physical inactivity was 93.1%, and was the most frequent risk behavior observed in almost all cancer survivors. However, there was a significant clustering pattern observed among female cancer survivors who were physically inactive but not obese, did not smoke, abstained from alcohol, and who frequently snacked in the ARM analysis. These important association rules demonstrated that physical inactivity co-occurred alongside relatively healthy behaviors, which in turn seems to have a preventive effect in logistic regression analysis [40]. Therefore, we demonstrated in our study that implementing ARM with a logistic regression analysis can aid in interpretation of the association among several risk factors in a hierarchy that frequently co-occur within the same subject.

The ARM analysis was firstly introduced to identify specific purchasing patterns of consumers in the market [26], and this is commonly known as a market basket analysis. The ARM has been shown to be useful for identifying unique and important features among a non-hierarchical data set. This analytic technique is also used in the non-economic data mining field [41,42]. We transformed all lifestyle risk behaviors into dichotomous variables for ARM analysis. As ARM was developed to be used with Boolean data, use of this method is restricted to studies that are designed to analyze associations between binary variables [43]. In addition, we identified health related behaviors in left-hand side (antecedent) and the ASCVD group in right-hand side (consequent) in our ARM analysis, which suggests that health related risks were the cause. However, ARM analysis is not designed to evaluate causal relationships, but to show coincidental associations, which requires additional attention in its application and interpretation [44].

Our study had some limitations. First, our results should be interpreted with caution, as this study was not a cohort study that directly compared the occurrence of cardiovascular disease. Instead, our study was an observational case–control study using the 10-year ASCVD score, which is a surrogate marker for cardiovascular disease. Second, this study analyzed the lifestyles risk factors among cancer survivors based on the KNHANES survey, which may not represent all cancer survivors because it is likely that cancer survivors with less severe cancers responded more commonly than those with more severe cancers. Nevertheless, our work included data on cancer survivors extracted from the nationally representative population, minimizing the selection bias resulting from a retrospective study design. Lastly, we did not assess the dietary habit of our participants. As a result, there may be a residual confounding bias between lifestyle risk behavior and cardiovascular disease risk. Actually, dietary habit is the potent lifestyle risk behavior for cancer and cardiovascular disease [45,46]. In addition, it has been confirmed in a large number of prospective cohorts that the detailed assessment of dietary habit such as plant-based dietary pattern is beneficial to differentiate high or low cardiovascular disease risk in the general population [47,48]. However, for the ARM method we used, it is difficult to interpret and could be sensitive to local association rules if we put many variables into the model. Therefore, we could not include a variety of variables in this analytical model, such as diet habit. Further research is needed on whether health risk behavior, which reflects the concurrent diet status, is associate with the ASCVD risk.

In summary, current smoking and obesity were identified as important lifestyle risk behaviors in overall cancer survivors. In addition, multiple lifestyle risk behaviors co-occurred with smoking in male cancer survivors. Health care providers should consider that targeted education and interventions to address lifestyle risk behaviors should prioritize their patients who have these selective risk clusters for behavioral intervention.

## Figures and Tables

**Figure 1 jpm-11-00366-f001:**
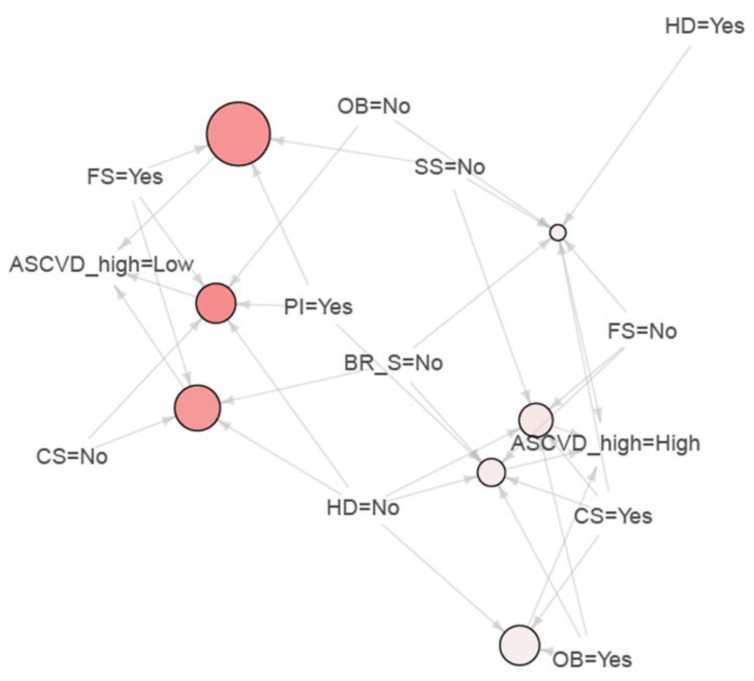
Association plot for male cancer survivors. Footnote: CS, current smoking; HD, heavy drinking; PI, physical inactivity; OB, obesity; SS, Suboptimal sleep; BR_S, breakfast skipping; FS, frequent snacking; ASCVD, atherosclerotic cardiovascular disease.

**Figure 2 jpm-11-00366-f002:**
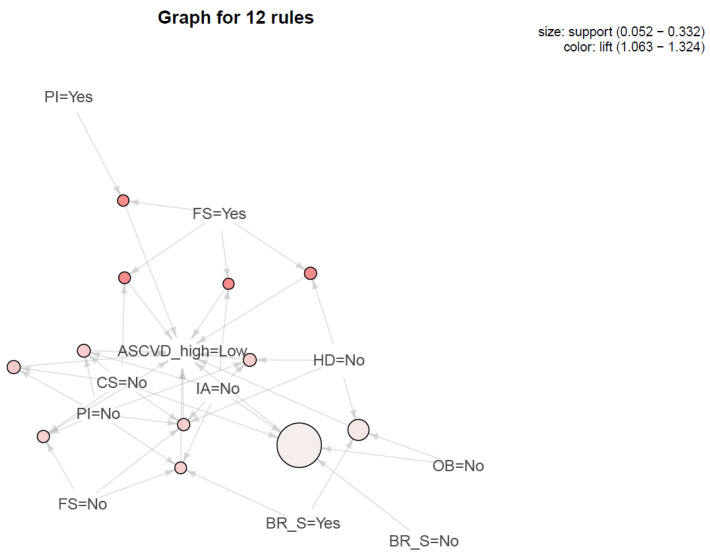
Association plot for female cancer survivors. CS, current smoking; HD, heavy drinking; PI, physical inactivity; OB, obesity; IS, inadequate sleep; BR_S, breakfast skipping; FS, frequent snacking; ASCVD, atherosclerotic cardiovascular disease.

**Table 1 jpm-11-00366-t001:** Comparison of baseline characteristics between low and high ASCVD score in cancer survivors.

	ASCVD Score			*p* Value
	Low (N = 520)	High (N = 377)	Total (N = 897)	
Age	54.8 ± 9.6	72.3 ± 5.6	62.2 ± 8.1	<0.001
Male	113 (21.7%)	245 (65.0%)	358 (39.9%)	<0.001
BMI, kg/m^2^	23.6 ± 3.2	23.8 ± 3.0	23.7 ± 3.1	0.267
Household income				0.368
Lowest	112 (21.5%)	98 (26.0%)	210 (23.4%)	
Lower middle	129 (24.8%)	94 (24.9%)	223 (24.9%)	
Upper middle	138 (26.5%)	85 (22.5%)	223 (24.9%)	
Highest	141 (27.2%)	100 (26.5%)	241 (27.0%)	
Educational year				<0.001
≤6 years	127 (24.4%)	197 (52.3%)	324 (36.1%)	
7–9 years	87 (16.7%)	48 (12.7%)	135 (15.1%)	
10–12 years	169 (32.5%)	68 (18.0%)	237 (26.4%)	
≥13 years	137 (26.4%)	64 (17.0%)	201 (22.4%)	
Marital status				<0.001
Yes	426 (81.9%)	271 (71.9%)	697 (77.7%)	
No	94 (18.1%)	106 (28.1%)	200 (22.3%)	
Hypertension	31 (6.0%)	32 (8.5%)	63 (7.0%)	0.184
Diabetes	4 (0.8%)	21 (5.6%)	25 (2.8%)	<0.001
Dyslipidemia				
Total cholesterol (mg/dL)	190.5 ± 36.6	182.3 ± 33.7	187 ± 35.6	0.001
HDL cholesterol (mg/dL)	52.0 ± 12.5	46.7 ± 11.8	49.8 ± 12.5	<0.001
LDL cholesterol (mg/dL)	113.8 ± 33.2	108.3 ± 31.1	111.5 ± 32.5	0.012
Blood pressure				
Systolic (mmHg)	115.8 ± 14.9	129.7 ± 16.5	121.6 ± 17.0	<0.001
Diastolic (mmHg)	74.9 ± 9.3	72.7 ± 9.7	74.0 ± 9.5	<0.001
Fasting blood glucose (mg/dL)	100.6 ± 25.1	107.4 ± 26.6	103.5 ± 25.9	<0.001
Alameda’s heath risk behavior				
Current smoking	31 (6.0%)	55 (14.6%)	86 (9.6%)	<0.001
Heavy drinking	54 (10.4%)	54 (14.3%)	108 (12.0%)	0.092
Physical inactivity	474 (91.2%)	361 (95.8%)	835 (93.1%)	0.011
Obesity	146 (28.1%)	118 (31.3%)	264 (29.4%)	0.331
Suboptimal sleep	68 (13.1%)	64 (17.0%)	132 (14.7%)	0.126
Breakfast skipping	142 (27.3%)	87 (23.1%)	229 (25.5%)	0.175
Frequent snacking	53 (10.2%)	5 (1.3%)	58 (6.5%)	<0.001

BMI, body mass index; HDL, high-density lipoprotein; LDL, low-density lipoprotein.

**Table 2 jpm-11-00366-t002:** Results of binary logistic regression analysis for predictors of high ASCVD score in cancer survivors.

	Crude Model	Model 1	Model 2	Model 3
	OR (95% CI)			
Current smoking	2.90 (1.77–4.75)	11.79 (3.82–36.37)	11.85 (3.84–36.51)	11.19 (3.66–34.20)
Heavy drinking	1.14 (0.75–1.75)	2.84 (1.02–7.88)	2.85 (1.02–7.92)	2.79 (0.99–7.85)
Physical inactivity	1.94 (1.02–3.71)	0.26 (0.07–0.94)	0.26 (0.07–0.97)	0.23 (0.06–0.86)
Obesity	1.12 (0.83–1.52)	2.81 (1.49–5.32)	2.84 (1.50–5.38)	2.67 (1.40–5.08)
Suboptimal sleep	1.44 (0.98–2.13)	2.02 (0.87–4.70)	2.06 (0.88–4.82)	1.90 (0.79–4.57)
Breakfast skipping	0.94 (0.66–1.33)	1.14 (0.56–2.30)	1.15 (0.57–2.32)	1.12 (0.55–2.27)
Frequent snacking	0.11 (0.04–0.27)	0.57 (0.13–2.56)	0.59 (0.13–2.67)	0.54 (0.12–2.45)

Model 1: crude model with adjusting for age and sex. Model 2: model 1 with adjusting for educational year. Model 3: model 2 with adjusting for household income and marital status.

**Table 3 jpm-11-00366-t003:** Results of a priori algorithms of for ASCVD score in male cancer survivors.

	LHS	RHS	Support	Confidence	Lift	Count
1	{SS = No, FS = Yes}	{ASCVD_high = Low}	0.052	1.000	1.324	28
2	{PI = Yes, FS = Yes}	{ASCVD_high = Low}	0.054	1.000	1.324	29
3	{HD = No, FS = Yes}	{ASCVD_high = Low}	0.061	1.000	1.324	33
4	{CS = No, FS = Yes}	{ASCVD_high = Low}	0.058	1.000	1.324	31
5	{CS = No, PI = No}	{ASCVD_high = Low}	0.069	0.902	1.195	37
6	{HD = No, PI = No, IA = No}	{ASCVD_high = Low}	0.067	0.900	1.192	36
7	{CS = No, PI = No, IA = No}	{ASCVD_high = Low}	0.067	0.900	1.192	36
8	{CS = No, PI = No, FS = No}	{ASCVD_high = Low}	0.061	0.892	1.181	33
9	{CS = No, HD = No, PI = No, IA = No, FS = No}	{ASCVD_high = Low}	0.061	0.892	1.181	33
10	{PI = No, IA = No, BS = Yes, FS = No}	{ASCVD_high = Low}	0.058	0.886	1.173	31
11	{HD = No, OB = No, BS = Yes}	{ASCVD_high = Low}	0.135	0.820	1.086	73
12	{CS = No, OB = No, IA = No, BS = No}	{ASCVD_high = Low}	0.332	0.803	1.063	179

Abbreviation: LHS = left hand side, RHS = right hand side, CS = current smoking, HD = heavy drinking, PI = physical inactivity, OB = obesity, FS = frequent snacking, SS = Suboptimal sleep, BS = breakfast skipping.

**Table 4 jpm-11-00366-t004:** Results of a priori algorithms of for ASCVD score in female cancer survivors.

	LHS	RHS	Support	Confidence	Lift	Count
1	{SS = No, FS = Yes}	{ASCVD_high = Low}	0.052	1.000	1.324	28
2	{PI = Yes, FS = Yes}	{ASCVD_high = Low}	0.054	1.000	1.324	29
3	{HD = No, FS = Yes}	{ASCVD_high = Low}	0.061	1.000	1.324	33
4	{CS = No, FS = Yes}	{ASCVD_high = Low}	0.058	1.000	1.324	31
5	{CS = No, PI = No}	{ASCVD_high = Low}	0.069	0.902	1.195	37
6	{HD = No, PI = No, SS = No}	{ASCVD_high = Low}	0.067	0.900	1.192	36
7	{CS = No, PI = No, SS = No}	{ASCVD_high = Low}	0.067	0.900	1.192	36
8	{CS = No, PI = No, FS = No}	{ASCVD_high = Low}	0.061	0.892	1.181	33
9	{CS = No, HD = No, PI = No, SS = No, FS = No}	{ASCVD_high = Low}	0.061	0.892	1.181	33
10	{PI = No, SS = No, BS = Yes, FS = No}	{ASCVD_high = Low}	0.058	0.886	1.173	31
11	{HD = No, OB = No, BS = Yes}	{ASCVD_high = Low}	0.135	0.820	1.086	73
12	{CS = No, OB = No, SS = No, BS = No}	{ASCVD_high = Low}	0.332	0.803	1.063	179

Abbreviation: LHS = left hand side, RHS = right hand side, CS=Current smoking, HD=Heavy drinking, PI=Physical inactivity, OB=Obesity, SS=Suboptimal sleep, BR_S=Breakfast skipping, FS=Frequent snacking.

## Data Availability

Publicly available datasets were analyzed in this study. The data can be found here: http://knhanes.cdc.go.kr; (Accessed date: 7 September 2020).

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
