# Peer review of "An Association Rule Mining Analysis of Lifestyle Behavioral Risk Factors in Cancer Survivors with High Cardiovascular Disease Risk"

_jpm, 2021, doi:10.3390/jpm11050366_

Round 1

Reviewer 1 Report

Although the information is interesting, and explores a new population, the foundation of the study needs updated.  Two key references date from 1974 and 2003.  Much has been learned about lifestyle that needs to be included in an evaluation of lifestyle related risk factors.  The Western lifestyle has, in addition become more toxic, and its impact on health has become even more marked than described in those studies.  A simple perusal of the American College of Lifestyle Medicine website will provide ample recent evidence on lifestyle and its relation to cancer and cardiovascular disease: Scientific Evidence (lifestylemedicine.org)

Diet is arguably the most potent lifestyle risk factor for cancer and cardiovascular disease: a plant-based, whole food diet being the only treat modality that has been shown to actually reverse cardiovascular disease (Updating a 12-year experience with arrest and reversal therapy for coronary heart disease (an overdue requiem for palliative cardiology) - PubMed (nih.gov)).  Even though the study mentions dietary information being available, the actual dietary components are not explored, but rather  "obesity" is utilized as a dietary surrogate.

Assuming the dietary information is adequate, a much better evaluation could be done by exploring dietary factors like fruit and vegetable, red and processed meat, and dairy consumption, etc.  Studies such as these would be good references / maybe a template:   Healthful and unhealthful plant-based diets and the risk of coronary heart disease in US adults (nih.gov),

Plant-Based Dietary Patterns and Incidence of Type 2 Diabetes in US Men and Women: Results from Three Prospective Cohort Studies (nih.gov)

Finding that smoking and obesity are linked to cardiovascular disease in cancer survivors really is not anything new.  Maybe it is in this population.  However, if you make the extra effort to explore the dietary factors, I think you will find something even more interesting and newsworthy, that may actually provide a new avenue of therapeutic focus in this population!

Author Response

Reviewer #1.

Although the information is interesting, and explores a new population, the foundation of the study needs updated. Two key references date from 1974 and 2003. Much has been learned about lifestyle that needs to be included in an evaluation of lifestyle related risk factors. The Western lifestyle has, in addition become more toxic, and its impact on health has become even more marked than described in those studies. A simple perusal of the American College of Lifestyle Medicine website will provide ample recent evidence on lifestyle and its relation to cancer and cardiovascular disease: Scientific Evidence (lifestylemedicine.org)

Diet is arguably the most potent lifestyle risk factor for cancer and cardiovascular disease: a plant-based, whole food diet being the only treat modality that has been shown to actually reverse cardiovascular disease (Updating a 12-year experience with arrest and reversal therapy for coronary heart disease (an overdue requiem for palliative cardiology) - PubMed (nih.gov)). Even though the study mentions dietary information being available, the actual dietary components are not explored, but rather "obesity" is utilized as a dietary surrogate.

Assuming the dietary information is adequate, a much better evaluation could be done by exploring dietary factors like fruit and vegetable, red and processed meat, and dairy consumption, etc. Studies such as these would be good references / maybe a template: Healthful and unhealthful plant-based diets and the risk of coronary heart disease in US adults (nih.gov),

Plant-Based Dietary Patterns and Incidence of Type 2 Diabetes in US Men and Women: Results from Three Prospective Cohort Studies (nih.gov)

Finding that smoking and obesity are linked to cardiovascular disease in cancer survivors really is not anything new.  Maybe it is in this population.  However, if you make the extra effort to explore the dietary factors, I think you will find something even more interesting and newsworthy, that may actually provide a new avenue of therapeutic focus in this population! 

: We totally agreed with that healthy diet is an important measure to prevent the individual’s cardiovascular disease in general population as well as in cancer survivors. However, for the ARM method we used, it is difficult to interpret and could be sensitive to local association rules if we put many variables into the model. Therefore, we could not include a variety of variables in this analytical model, such as diet habit. We further describe the limitation of the study, citing two papers presented by the reviewer that the diet information is important factor in assessing CVD risk in general population. For topics such as the healthful plant-based diet index calculated in the presented paper, we will conduct further research in collaboration with the nutrition specialist later.

Reviewer 2 Report

In the crude model in Table 2, the 95% Cl of obesity and heavy drinking cover 1. This means that these two factors are not significant. In particular, the authors claim that obesity is an important lifestyle risk behavior among cancer survivors. Only adjusted models support this claim. It should be addressed in the article.

Author Response

see attached file for response to reviewer 2

Round 2

Reviewer 1 Report

Thank you for the changes made.  I still believe that the most important variable you could have looked at would have been dietary composition.  I do understand the difficulty in doing this, however, it would have added greatly to therapeutic options if you demonstrated, for example, that red or processed meat consumption more than three times a week vs less than three times a week was a risk factor.